

# Spatial near future modeling of land use and land cover changes in the temperate forests of Mexico

Jesús A. Prieto-Amparán[1], Federico Villarreal-Guerrero[1], Martin Martínez-Salvador[1], Carlos Manjarrez-Domínguez[2], Griselda Vázquez-Quintero[2] and Alfredo Pinedo-Alvarez[1]

[1] Facultad de Zootecnia y Ecología, Universidad Autónoma de Chihuahua, Chihuahua, Chihuahua, Mexico
[2] Facultad de Ciencias Agrotecnológicas, Universidad Autónoma de Chihuahua, Chihuahua, Chihuahua, Mexico

Corresponding author
Alfredo Pinedo-Alvarez,
apinedo@uach.mx

## ABSTRACT

The loss of temperate forests of Mexico has continued in recent decades despite wide recognition of their importance to maintaining biodiversity. This study analyzes land use/land cover change scenarios, using satellite images from the Landsat sensor. Images corresponded to the years 1990, 2005 and 2017. The scenarios were applied for the temperate forests with the aim of getting a better understanding of the patterns in land use/land cover changes. The Support Vector Machine (SVM) multispectral classification technique served to determine the land use/land cover types, which were validated through the Kappa Index. For the simulation of land use/land cover dynamics, a model developed in Dinamica-EGO was used, which uses stochastic models of Markov Chains, Cellular Automata and Weight of Evidences. For the study, a stationary, an optimistic and a pessimistic scenario were proposed. The projections based on the three scenarios were simulated for the year 2050. Five types of land use/land cover were identified and evaluated. They were primary forest, secondary forest, human settlements, areas without vegetation and water bodies. Results from the land use/land cover change analysis show a substantial gain for the secondary forest. The surface area of the primary forest was reduced from 55.8% in 1990 to 37.7% in 2017. Moreover, the three projected scenarios estimate further losses of the surface are for the primary forest, especially under the stationary and pessimistic scenarios. This highlights the importance and probably urgent implementation of conservation and protection measures to preserve these ecosystems and their services. Based on the accuracy obtained and on the models generated, results from these methodologies can serve as a decision tool to contribute to the sustainable management of the natural resources of a region.

## INTRODUCTION

Forest ecosystems are important because they provide a wide variety of products and services for the human well being (*Hall et al., 2006*; *Fischer & Lindenmayer, 2007*; *Weiskittel, Crookston & Radtke, 2011*) harvested products (*Houghton & Nassikas, 2017*),

carbon sequestration (*Hawkes et al., 2017*), soil retention (*Borrelli et al., 2017*), water supply (*Sun et al., 2006*) and are the habitat of many species of plants and animals. However, antrophongenic activities are the main cause of degradation of almost half of the world surface in the last three centuries. That has caused the loss of lots of our precious natural resources. Twenty-five nations have practically degraded 100% of their forests, and another 29 nations have degraded 10% of their forest areas (*Millennium Ecosystem Assessment, 2005*).

Temperate forests represent a key element in the carbon cycle (*Pan et al., 2011*). They are important carbon dioxide sinks (*Ma, Jia & Zhang, 2017*), offsetting the emissions produced by the world population (*FAO, 2018*). Temperate forests store 14% of the planet's carbon (*Pan et al., 2011*). However, projections of global environmental change show that temperate forests show high vulnerability (*Gonzalez et al., 2010*). This vulnerability can change the productivity of forests by modifying net carbon sequestration rates (*Peters et al., 2013*).

Temperate forests of Mexico occupy 17% of the national territory, represented by 32 millions hectares. In this region, the greatest association of pine and oak forests in the world occurs (*González, 2012*). Around 23 different species of pines and close to 200 species of oaks live in the ecoregion of Sierra Madre Occidental (*Navar, 2009*). However, 40 thousand hectares of forests get on average lost annually. This region has the highest deforestation rate in the world (*Velázquez et al., 2002*; *Mas et al., 2004*).

The study of the land use/land cover changes (LULCC) has become a fundamental research topic, since the change in land use/land cover (LULC) affects forest ecosystems and their biodiversity (*Gharun et al., 2017*). The LULCC, produced by anthropogenic activities have significantly altered the ecosystems biodiversity and services (*Butler & Laurance, 2008*; *Miles & Kapos, 2008*; *Miranda-Aragón et al., 2012*). The dynamics of LULCC directly affect the landscape patterns, the biogeochemical cycles, the ecosistems structure and function (*Scheffer et al., 2001*). Recently, the analysis of the spatio-temporal patterns has been the objective of several research studies (*Huang, Zhang & Wu, 2009*; *Manjarrez-Dominguez et al., 2015*; *Vázquez-Quintero et al., 2016*). The models of LULCC commonly employed, quantify deforested surfaces, measuring the degree of change in the ecosystem (*Lapola et al., 2011*). Regression methods suchs as the logistic regression have been employed to generate models of LULCC. These models suppose that the relationship between the LULCC and the variables that produce it is a logistic function; however, it has been demonstrated that this relationship is too general (*Mas et al., 2010*; *Mas et al., 2014*). The dynamics and complexity of the ecosystem requires a more complete evaluation of LULCC. The spatial modeling is a technique contemplating alternative scenarios of LULCC, which could contribute to better explain the key processes influencing LULCC (*Pijanowski et al., 2002*; *Eastman, Solorzano & Van Fossen, 2005*; *Torrens, 2006*; *Pérez-Vega, Mas & Ligmann-Zielinska, 2012*). Thus, one of the main functions of the LULCC models is the establishment of scenarios, with the aim of changing policies and inadequate practices for the sustainable management of natural resources (*DeFries et al., 2007*; *Berberoğlu, Akin & Clarke, 2016*).

Several approaches to establish LULCC scenarios have been developed and tested to generate scenarios of LULCC. *Ferreira et al. (2012)* generated deforestation scenarios to 2050 in the central Brazilian savanna biome finding the possible increase of 13.5% in deforested areas. *Kamusoko et al. (2011)* evaluated three scenarios (optimistic, pessimistic and business-as-usual) in the Luangprabang province, Lao People's Democratic Republic, finding decreases in forest areas in the pessimistic and business-as-usual scenarios and an increase in forest areas in the optimistic scenario under a strict regulatory policy. *Gago-Silva, Ray & Lehmann (2017)* used a combination of Bayesian methods and Weights of Evidence to model the probability of change in a western part of Switzerland. *Galford et al. (2015)* used Bayesian Weights of Evidence for policiy scenarios from 2010 a 2050 evaluating plans for agriculture and forest in Democratic Republic of Congo.

The models to establish reference scenarios of changes in LULCC are based on: systems of equations, statistic models, experts, evolutionary and cellular models, even though there have been efforts to combine plataforms in a multiagent system (*Mas et al., 2014*; *Stan & Sanchez-Azofeifa, 2017*). The statistical models employ spatial statistics and regression, in comparison with the expert models, which allow the expert knowledge to lead the model path (*Parker et al., 2003*; *Soares-Filho, Rodrigues & Follador, 2013*). The evolutionary or cellular models are very competent to determine the ecologycal alteration; however, they just provide information about the causality or the decision-making (*Parker et al., 2003*).

The generation of LULCC scenarios for the forest region of the state of Chihuahua, Mexico is necessary because of the higher temperate forest deforestation rates in the country. The generation of the LULCC scenario shows two important aspects: expert knowledge and knowledge based on data. Expert knowledge is useful to establish methodological processes according to the needs of the user (*Gounaridis, Chorianopoulos & Koukoulas, 2018*). Knowledge based on data, helps to understand the general behavior between the factors of change of land use in a spatial way (*Olmedo et al., 2018*). Most studies are based on knowledge of the data (*Paegelow & Olmedo, 2005*; *Kityuttachai et al., 2013*), however, few allow the inclusion of both (*Soares-Filho et al., 2006*; *Olmedo et al., 2018*).

The Dinamica Environment for Geoprocessing Objects (Dinamica-EGO) is a flexible open platform, which allows analyzing distribution, abundance and spatio-temporal dynamic of the landscape (*Soares-Filho, Cerqueira & Pennachin, 2002*; *Lima et al., 2013*). The model incorporated to Dinamica-EGO employs cellular automata to simulate the changes happening in a grid, estimating the transition probability, as well as the direction of changes based in stocastic processes (*Rutherford et al., 2008*; *Arsanjani et al., 2013*). Dinamica-EGO allows users to incorporate expert knowledge into the overall statistical analysis based on the spatial data set (*Mas et al., 2014*). In addition, Dinamica-EGO incorporates the possibility of modifying landscape metrics in the calibration procedure to generate the simulation (*Mas, Pérez-Vega & Clarke, 2012*). In a comparative evaluation of approaches to modeling LULCC, two key advantages over Dinamica-EGO were emphasized: (1) incorporation of the Patcher and Expander functions. The first function generates new patches in the landscape and the second expands the previously formed patches, (2) Dinamica-EGO allows the incorporation of multiresolution validation by means of the Fuzzy Similarity Index.

The aim of the present study was (a) to evaluate the change dynamics in the period from 1990 to 2017; (b) to simulate the changes of LULCC for the year 2050 and (c) to elaborate a discussion about the impacts of different scenarios, which could happen in the future in a forest region of the state of Chihuahua, Mexico. Specifically, there are three scenarios, pessimistic, optimistic and stationary state. The model will identify where the different types fo LULCC could hapen. This will allow that future studies could determine changes in carbon sequestration in both, on the surface extension and quantity.

## MATERIALS & METHODS

### Study area

The study area is located in the western part of the state of Chihuahua, Mexico. It is part of the 'Sierra Tarahumara' and have a surface area of 497,159 ha. Its extreme coordinates are 108°00′W, 29°00′N and 107°10′W, 27°30′N (Fig. 1). It is one of the regions of temperate forests, which has experimented the greatest disturbances in the past years in the state of Chihuahua (Herrera, 2002). It belongs to the most extensive forest areas in North America. It is immersed within a complex orography composed of large canyons and deep canyons, which results in a mixture of temperate and tropical ecosystems. It is characterized by its high biodiversity and number of endemic species, estimating the presence of around 4,000 species of plants. Also, it is recognized by the International Union for the Conservation of Nature as one of the megacenters of plant diversity (Felger & Wilson, 1995). The main land uses in the area include: pine forests, oak forests, pine-oak and oak-pine forest associations, agriculture and grassland communities. The economic activities in the region are forestry, extensive livestock and rainfed agriculture (INEGI, 2003).

### Data source

For the analysis of the LULCC, three scenes of the Landsat sensor (Path 33, Row 41), with a spatial resolution of 30 m, were used. The scenes corresponded to the years 1990, 2005 and 2017 and they were acquired from clear sky days and each of them taken during the same month to reduce the temporal variation. The scenes were downloaded from the United States Geological Survey (USGS, 2018). The characteristics of each scene can be seen in Table 1.

The scenes were radiometrically corrected. The radiometric correction was carried out with the QGis software v.2.8 through the SemiAutomatic Classification plugin (Congedo, 2017).

### Integration and composition of bands

Once the scenes were corrected, they were integrated into a layer stack. False color composites for the Landsat TM5 were then generated, with a combination of the bands 5, 4 and 3. Band 5 corresponds to the infrared channel (1.55–1.75 µm), band 4 to the near infrared (0.76–0.90 µm) and the band 3 to the red channel (0.63–0.69 µm). This combination was applied to the scenes of 1990 and 2005. Regarding the scene of 2017, the combination for Landsat OLI8 was applied and corresponded to the bands 6, 5 and 4, where band 6 corresponds to the medium infrared channel (1.55–1.65 µm), band 5 to

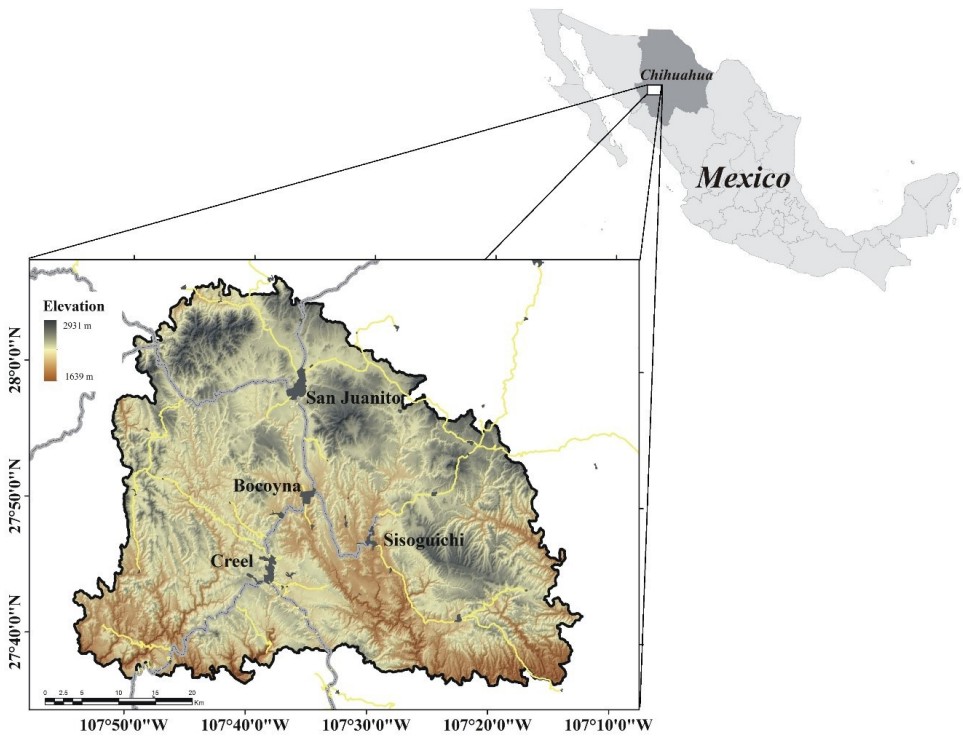

**Figure 1** Location and elevations of the study area.

**Table 1** Scenes characteristics.

| Sensor | Date | Characteristics |
|---|---|---|
| Landsat TM 5 | 1990 | 7 spectral bands, 30 m resolution |
| Landsat TM 5 | 2005 | 7 spectral bands, 30 m resolution |
| Landsat OLI 8 | 2017 | 8 spectral bands, 30 m resolution; 1 panchromatic band 15 m resolution |

Notes.
TM, Thematic Mapper; OLI, Operational Land Imager.

the near infrared channel (0.85–0.88 μm) and band 4 to the red channel (0.64–0.67 mn) (*Lillesand & Kiefer, 2000*).

## Land use and land cover classification

The Suport Vector Machine (SVM) classification was applied to the 1990, 2005 and 2017 images through the software R (*R Core Team, 2016*) with the R package ''caret'' (*Kuhn, Wing & Weston, 2015*) to obtain LULC information. The SVM classifier is a supervised technique of nonparametric statistical methods (*Mountrakis, Im & Ogole, 2011*). The SVM classification has been used in several research studies in the past (*Kavzoglu & Colkesen, 2009*; *Otukei & Blaschke, 2010*; *Shao & Lunetta, 2012*). For the supervised classification, five classes of land use were defined; (1) primary forest, (2) secondary forest, (3) human settlements, (4) areas without vegetation and (5) water bodies (Table 2).

**Table 2  Land use/land cover types determined through the supervised classification method.**

| Land use and land cover | Acronym | Description |
|---|---|---|
| Primary forest | PF | Forest fully covered with canopy |
| Secondary forest | SF | Forest partially covered with canopy |
| Human settlements | HS | Residential areas |
| Areas without vegetation | AWV | Areas without vegetation, agriculture areas or induced grasslands |
| Water bodies | WB | Water bodies |

## Modeling and spatial simulation with Dinamica-EGO

The LULCC scenarios were made based on the historical trends of change in forest cover during 1990–2017 of the supervised classifications using Dinamica-EGO (*Soares-Filho, Cerqueira & Pennachin, 2002*). The historical trends of LULCC is based on the transition matrix (*Monteiro Junior, Silva E De Amorim Reis & Mesquita Souza Santos, 2018*). Dinamica-EGO uses the algorithm of cellular automata, and the method Weights of Evidence (*Olmedo et al., 2018*). For the simulation of deforestation, the following steps were undertaken: (1) selection of change drivers as well as transitions, (2) exploratory analysis of the drivers of deforestation, (3) simulation and (4) validation. These four steps are described in the following sections.

## Selection of variables and transitions

The selection of the set of exploratory variables to simulate the LULCC is essential for the modeling success (*Miranda-Aragón et al., 2012*; *Pérez-Vega, Rocha Álvarez & Regil García, 2016*). In this study, 19 variables were used; 17 static and two dynamic variables. Static variables remain constant during model execution. Dynamic variables change during the execution of the model and they are continuously updated in each iteration (*Olmedo et al., 2018*). The set of variables used is shown in Table 3.

The transition refers to the total amount of LULCC that occurred in the simulation period. In this study, the transitions of interest were: (a) primary forest to secondary forest, (b) primary forest to areas without apparent vegetation, (c) primary forest to urban areas and (d) secondary forest to areas without apparent vegetation (Table 4).

## Exploratory analysis of the data

When we modeled LULCC dynamics, Weights of Evidence (WoE) were applied to project transition probabilities. Regarding deforestation, degradation or any other type of change, we previously know about the location of favorable conditions for LULCC. The influence of static and dynamic variables and the elaboration of the LULC maps was performed with WoE in the Dinamica-EGO software (*Soares-Filho et al., 2010*).

The positive values of WoE represent an attraction between a transition of land use and a specific variable. The greater the value of $W^+$, the greater the probability of transition. Negative values of $W^-$ indicate low probabilities of transition instead (*Maeda et al., 2010a*; *Maeda et al., 2010b*). By using the WoE values of the variables used in the analysis of LULCC, the Dinamica-EGO model calculates the transition probability of each pixel to

**Table 3  Variables feeding the deforestation model.**

| No | Variable type | Name | Unit | Acronym |
|---|---|---|---|---|
| 1 | | Density of main roads | $m^2/Km^2$ | Denmr |
| 2 | | Density of secondary roads | $m^2/Km^2$ | Densr |
| 3 | Density | Density of main streams | $m^2/Km^2$ | Denms |
| 4 | | Density of secondary streams | $m^2/Km^2$ | Denss |
| 5 | | Density of rural settlements | $m^2/Km^2$ | Denrs |
| 6 | | Distance to sawmills | m | Diss |
| 7 | | Distance to water bodies | m | Diswb |
| 8 | | Distance to main roads | m | Dismr |
| 9 | | Distance to secondary roads | m | Dissr |
| 10 | Proximity | Distance to main streams | m | Disms |
| 11 | | Distance to secondary streams | m | Disss |
| 12 | | Distance to rural settlements | m | Disrs |
| 13 | | Distance to urban settlements | m | Disus |
| 14 | | Distance to mines | m | Dism |
| 15 | | Distance to areas without apparent vegetation | m | Disawav |
| 17 | | Altitude | m | Alt |
| 18 | Topographic | Slope | ° | Slop |
| 19 | | Topographic position index | Dimensionless | TPI |

**Table 4  Transitions of land use/land cover.**

| | | To | | | | |
|---|---|---|---|---|---|---|
| | | **PF** | **SF** | **HS** | **AWV** | **WB** |
| | PF | | ✓ | ✓ | ✓ | |
| | SF | | | | ✓ | |
| **From** | HS | | | | | |
| | AWV | | | | | |
| | WB | | | | | |

change. Thus, the pixels are assigned with a probability value for a given transition and probability maps are generated for the transitions of interest (*Soares-Filho, Rodrigues & Costa, 2009*; *Soares-Filho et al., 2010*; *Mas & Flamenco, 2011*).

Given that the basic hypothesis of the WoE technique is that the driving variables must be independent, for this study the correlation between the variables was tested through the Cramer Coefficient ($V$), represented by Eq. (1).

$$V = \sqrt{\frac{\chi^2}{\Gamma\ldots M}} \tag{1}$$

Where: $\chi^2 =$ is the chi-square statistic of the contingency between two variables, $\Gamma =$ denotes the sum of the values of contingency, $M =$ is the minimum of $n-1$ or $m-1$, where $n$ denotes the number of rows and $m$ the number of columns. *Bonham-Carter (1994)*

mentioned that values lower than 0.5 for the Cramer Coefficient ($V$) suggest independence, while values higher than 0.5 involve a greater association (*Almeida et al., 2003*; *Teixeira et al., 2009*).

## Simulation of land use and land cover changes

Three types of scenarios were used for 2050; they were called pessimistic, optimistic and stationary. For the three scenarios, the modeling base was the period 1990–2017. The transition matrix of 1990 and 2017 were used to estimate the possible change in forestry coverage in the future, taking 2017 as the beginning year and 2050 as the final year. In the pessimistic scenario, the transition probability matrix and the change function (patcher and expander) were modified, increasing the deforestation and fragmentation rates between 1990 and 2017. This was done based on the hypothesis that the development of road infrastructure, urban expansion, fires, uncontrolled exploitation, among others, will produce strong spatial changes of land use. For the optimistic scenario, the state and national forest development plans were considered. Such plans promote the protection and conservation of forest resources (*CONAFOR, 2001*). For this scenario, the conservation and promotion of strategies to protect forests were represented by reducing the transition matrix value, as well as the patcher and expander change functions. Regarding the stationary scenario, transitions or change functions were not modified. In this case, it is assumed that the trend will be the same as the one between 1990 and 2017.

## Validation

To evaluate the model performance, we used a Fuzzy Similarity Index (FSI), where the representation of a pixel is influenced by itself and its neighborhood (*Ximenes et al., 2011*; *Yanai et al., 2012*; *Chadid et al., 2015*). The FSI employed in this study was developed by *Hagen (2003)*, modified by *Soares-Filho et al. (2017)* and implemented in Dinamica-EGO. The FSI verifies the agreement between the observed and the simulated land use and land cover datasets by obtaining the number of coincident cells within increasing window sizes of a neighborhood (*Costanza, 1989*; *Soares-Filho et al., 2017*). The validation process was carried out by comparing a simulated map and a reference map. The simulation of the 2017 LULCC map was generated. To generate the simulation of 2017, the transition matrix was used between 1990 and 2005. The comparison through the FSI allowed to evaluate the areas of coincidence of change and no change between the real and simulated map of 2017. Finally, the general procedure used in this study is outlined in the flowchart depicted in Fig. 2.

# RESULTS

## Detection of land use/land cover changes

Results from the analysis of LULCC show a considerable gain for secondary forest. The forest cover of the primary forest was reduced from 55.8% of the study area in 1990 to 37.7% in 2017. The areas without vegetation increased their area from 4.11% to 4.87% during 1990–2017 (Table 5). Regarding human settlements and water bodies, they showed a positive trend with an increase from 0.03% and 0.01 in 1990 to 0.1% and 0.03 in 2017,
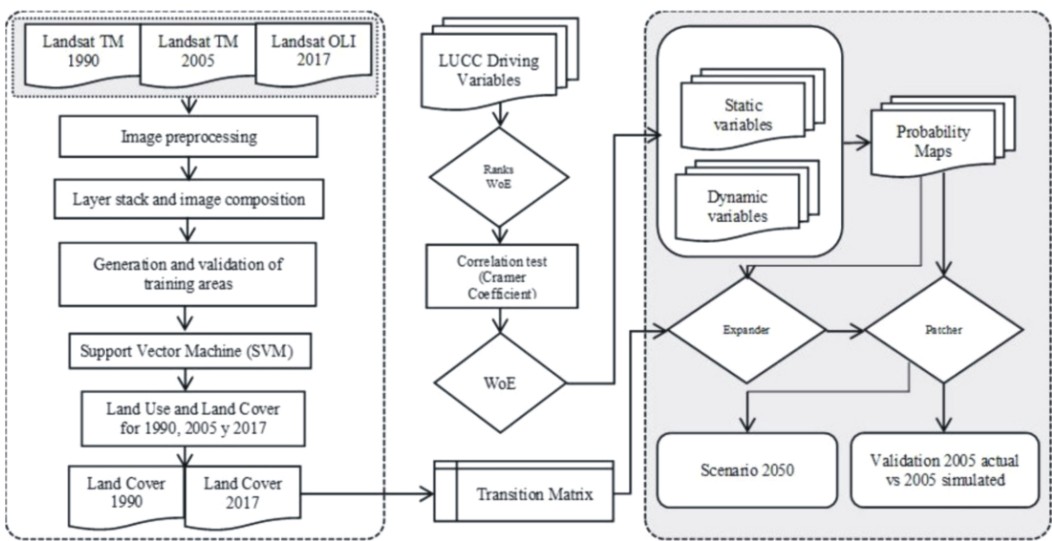

**Figure 2** Flowchart of the methodological procedure followed to produce the proposed scenarios. Abbreviations: TM, Tematic Mapper; OLI, Operational Land Imager; WoE, Weights of Evidence; LUCC, Land use and cover change.

**Table 5** Area occupied for five types of land uses during 1990, 2005 and 2017, and rate of change for the periods 1990–2005 and 2005–2017.

| Land Use | Occupied area (Ha) | | | Occupied area (%) | | | Rate of change | |
|---|---|---|---|---|---|---|---|---|
| | 1990 | 2005 | 2017 | 1990 | 2005 | 2017 | 1990–2005 | 2005–2017 |
| AWV | 20444.18 | 23828.59 | 24101.92 | 4.11 | 4.79 | 4.85 | 8.33 | 8.43 |
| SF | 199121.38 | 223948.16 | 286922.04 | 40.05 | 45.05 | 57.72 | 8.03 | 10.68 |
| HS | 154.60 | 272.35 | 521.65 | 0.03 | 0.05 | 0.10 | 12.58 | 15.96 |
| WB | 26.9712 | 103.76 | 153.91 | 0.01 | 0.02 | 0.03 | 27.48 | 12.36 |
| PF | 277380.46 | 248973.97 | 185427.79 | 55.80 | 50.08 | 37.30 | 6.41 | 6.21 |

**Notes.**

AWV, areas without vegetation; SF, secondary forest; HS, human settlements; WB, water bodies; PF, primary forest.

respectively. In general, the primary forest was the land use that experimented a negative trend. The rest of the land uses showed surface gains. The rate of change obtained indicate that the secondary forest, the human settlements and the water bodies were the land uses with the greatest transformation rates, with 8.03, 12.58 and 27.48, respectively, for the period of 1990–2017 and with 10.68, 15.96 and 12.3, respectively, from 2005 to 2017. Figure 3 shows the area occupied by the land uses studied. Likewise, it shows the rate of change of these land use/land cover for the periods 1990–2005 and 2005–2017. The calculated global precision, based on the Kappa Index, presented values of 80%, 85% and 84% for 1990, 2005 and 2017, respectively.

Table 6 shows the land use/land cover change dynamics. The primary forest lost the greatest surface area (28,406 ha) during 1990–2005, increasing the surface lost to 63,546 ha during 2005–2017. In contrast, the secondary forest showed the largest increases in area with 87,800 ha in the period 1990–2017.

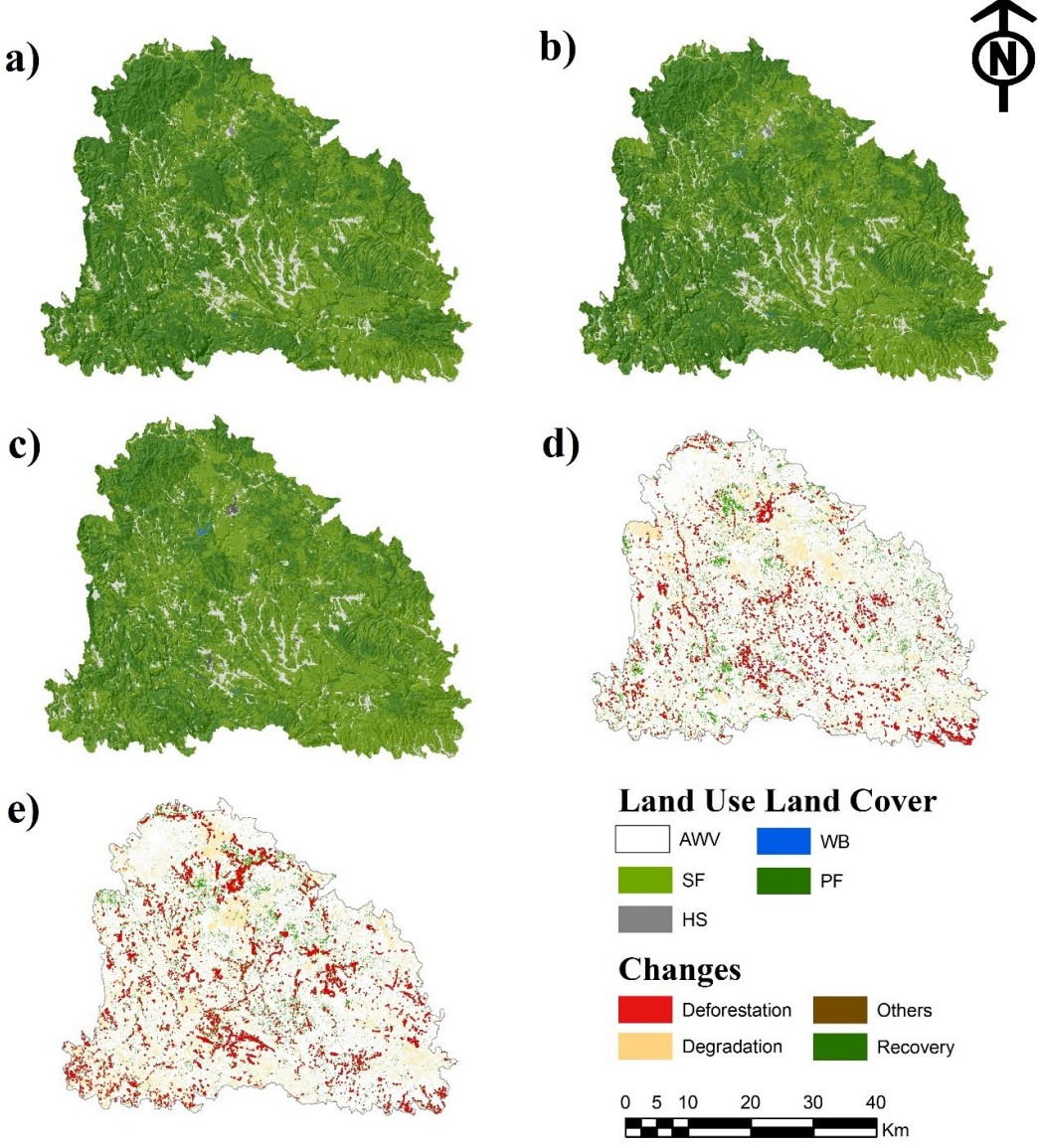

**Figure 3** Land use/land cover of 1990 (A), 2005 (B), 2017 (C), changes during 1990–2005 (D) and changes during 2005–2017. Abbreviations: AWV, areas without vegetation; SF, secondary forest; WB, water bodies; HS, human settlements and PF, primary forest.

## Transition matrix

The transition probabilities of LULCC for the periods 1990–2005 and 2005–2017 are shown in Table 7. The diagonal of the matrix represents the permanence probability, i.e., the probability of a LULC type to remain unchanged. The areas without vegetation showed a 90% probability of transition from 1990 to 2005, lowering it to 62% from 2005 to 2017. The areas of primary forest presented a negative trend with a 71% probability of permanence in the period 1990 to 2005, and changing it to 61% for the period 2005–2017.
**Table 6  Land use/land cover change dynamics.**

| Land use | Difference 1990–2005 (ha) | Difference 2005–2017(ha) | Overall Difference (ha) | Type of change | 1990–2005 (ha) | 2005–2017 (ha) |
|---|---|---|---|---|---|---|
| AWV | 3384.40 | 273.34 | 3657.74 | Deforestation | 3120.40 | 7283.95 |
| SF | 24826.78 | 62973.88 | 87800.66 | Degradation | 54455.78 | 73904.27 |
| HS | 117.74 | 249.31 | 367.05 | Other | 76.22 | 219.41 |
| WB | 76.79 | 50.15 | 126.94 | Recovery | 27128.71 | 20204.13 |
| PF | −28406.49 | −63546.18 | −91952.66 | – | – | – |

Notes.
  AWV, areas without vegetation; SF, secondary forest; HS, human settlements; WB, water bodies; PF, primary forest.

**Table 7  Transition matrix of probability for land use/land cover change (1990–2005, 2005–2017, 1990–2017).**

| | Periodo | AWV | PF | HS | WB | PF |
|---|---|---|---|---|---|---|
| | 1990–2005 | **0.9000** | 0.0250 | 0.0250 | 0.0250 | 0.0250 |
| AWV | 2005–2017 | **0.6250** | 0.3504 | 0.0108 | 0.0029 | 0.0109 |
| | 1990–2017 | **0.6615** | 0.3124 | 0.0120 | 0.0035 | 0.0106 |
| | 1990-2005 | 0.0222 | **0.7516** | 0.0008 | 0.0005 | 0.2248 |
| SF | 2005–2017 | 0.0557 | **0.8116** | 0.0004 | 0.0000 | 0.1323 |
| | 1990–2017 | 0.0654 | **0.7945** | 0.0012 | 0.0006 | 0.1384 |
| | 1990-2005 | 0.0452 | 0.0645 | **0.8806** | 0.0000 | 0.0097 |
| HS | 2005–2017 | 0.0557 | 0.2479 | **0.6959** | 0.0000 | 0.0004 |
| | 1990–2017 | 0.0651 | 0.0774 | **0.8575** | 0.0000 | 0.0000 |
| | 1990–2005 | 0.0000 | 0.1254 | 0.0000 | **0.8553** | 0.0193 |
| WB | 2005–2017 | 0.0095 | 0.1684 | 0.0000 | **0.8030** | 0.0191 |
| | 1990–2017 | 0.0000 | 0.1868 | 0.0000 | **0.7957** | 0.0175 |
| | 1990–2005 | 0.0020 | 0.2865 | 0.0000 | 0.0000 | **0.7115** |
| PF | 2005–2017 | 0.0056 | 0.3798 | 0.0003 | 0.0000 | **0.6144** |
| | 1990–2017 | 0.0071 | 0.4419 | 0.0002 | 0.0000 | **0.5508** |

Notes.
  AWV, areas without vegetation; SF, secondary forest; HS, human settlements; WB, water bodies; PF, primary forest.

## Weights of evidence (WoE) analysis

The WoE of the 19 variables were analyzed to eliminate those values that were above 0.5, based on the Cramer Coefficient ($V$). The distance to urban locations showed positive values of WoE from 1,000 to 9,000 m distance and from 42,000 to 47,000 m indicating an influence for cover change from secondary forest to area without vegetation. The distance to rural localities showed positive values of WoE in distances from 0 to 700 m. The topographic position index showed positive values in the ranges of −150 to −60 and 120 to 240. The distance to sawmills indicates that deforestation appears from 0 to 16,000 m with respect to the process of change between secondary forest to areas without vegetation. The transition from primary forest to area without vegetation is likely to occur in distances to the main roads between 13,000 and 21,000 m. The density of main streams such as rivers and creeks had an influence in densities from 0.039 to 0.079 m$^2$/km$^2$.

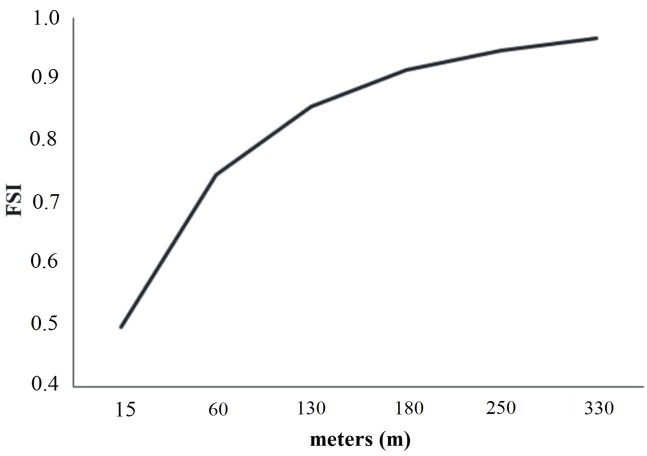

**Figure 4** Variation of the FSI as a function of different distance.

In the transition from primary forest to secondary forest, the variable altitude showed positive values of WoE in the range of 1,200–1,300 m, suggesting that most of the changes occur in this range. The slope showed that the process of change between primary forest and secondary forest is located on slopes of 45–60 and 60–75 degrees. The transition from primary forest to human settlements was influenced by the distance to secondary streams from 500 to 1,000 m. The distance to sawmills presented an influence from 0 to 6,000 m. The distance to mines showed that the attraction to change occurs between 2,000 and 10,000 m.

### Model validation

The model validation was carried with the simulated and the true land use classification of 2017. The FSI was applied for neighborhoods from $1 \times 1$ to $7 \times 7$ pixels. The minimum value reported for FSI was 49% in $1 \times 1$ pixels, while in $7 \times 7$ pixels the value of FSI was 91%. These results indicate that the real and simulated land use changes agree from 49% to 91%. Simulation starts with 49% and adjusts to 91%, reaching a similarity adjustment value at a distance of 210 m. These results agree with that obtained by *Ximenes et al. (2011)*. According to *Soares-Filho et al. (2017)*, and similar studies (*Carlson et al., 2012*; *De Rezende et al., 2015*; *Elz et al., 2015*), for the resolution and the number of transitions considered in the model, the values obtained for the FSI suggest that the models are good and can be used in the simulation of LULCC scenarios. Figure 4 represents the FSI in relation to the size of the window.

### Scenarios

The LULCC based on the transitions between 1990 and 2017 for the stationary, optimistic and pessimistic scenarios are presented in Table 8.

Figure 5 shows the LULC classification of 2017 and the stationary, optimistic and pessimistic scenarios for 2050, after the model calibration.

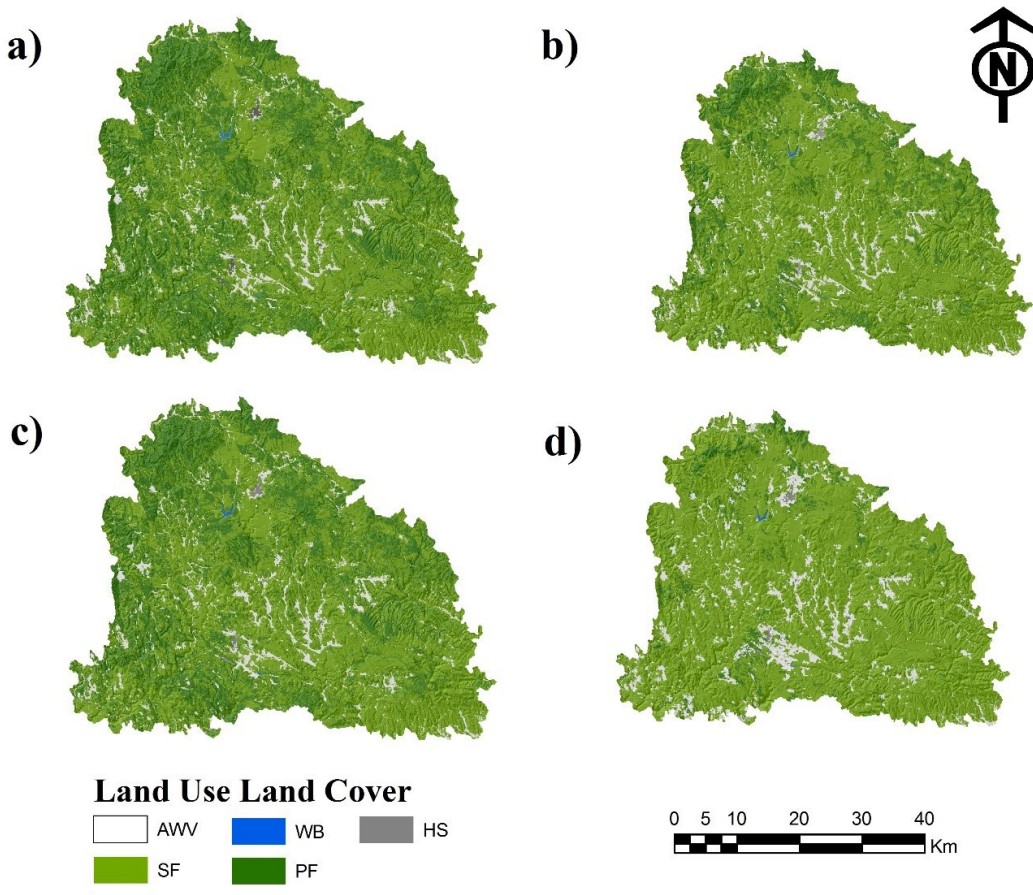

**Figure 5** (A) Land use/land cover of 2017 and simulated land use/land cover projected for the year 2050 as a result of the (B) Stationary, (C) Pessimistic and (D) Optimistic scenarios. Abbreviations: AWV, areas without vegetation; SF, secondary forest; WB, water bodies.

**Table 8** Percentage of surface area occupied by five land use/land cover types and rate of change for 2017–2050 based on three scenarios.

| Land use | Occupied surface area (%) | | | | Change rate | | |
|---|---|---|---|---|---|---|---|
| | 2017 | 2050s | 2050o | 2050p | 2017–2050s | 2017–2050o | 2017–2050p |
| AWV | 4.848 | 5.275 | 5.017 | 7.695 | 3.40 | 3.23 | 4.96 |
| SF | 57.716 | 73.721 | 61.863 | 83.628 | 3.99 | 3.35 | 4.53 |
| HS | 0.105 | 0.105 | 0.105 | 0.105 | 3.13 | 3.13 | 3.13 |
| WB | 0.031 | 0.031 | 0.031 | 0.031 | 3.12 | 3.13 | 3.12 |
| PF | 37.300 | 20.868 | 32.983 | 8.541 | 1.75 | 2.76 | 0.72 |

**Notes.**

AWV, areas without vegetation; SF, secondary forest; HS, human settlements; WB, water bodies; PF, primary forest; S, stationary; O, optimistic; P, pessimistic.

**Table 9   Land use/land cover change dynamics (ha) under three proyected scenario.**

| Land use | 2017–2050$_s$ | 2017–2050$_o$ | 2017–2050$_p$ |
|---|---|---|---|
| AWV | 2121.97 | 840.44 | 14150.28 |
| SF | 79565.57 | 20617.02 | 128818.00 |
| HS | 0.87 | 1.25 | 1.05 |
| WB | 0.46 | 0.32 | 0.10 |
| PF | −81688.00 | −21459.04 | −142969.31 |

Notes.

AWV, areas without vegetation; SF, secondary forest; HS, human settlements; WB, water bodies; PF, primary forest; S, stationary; O, optimistic; P, pessimistic.

In the stationary scenario the area without vegetation would increase from 4.8% in 2017 to 5.27%. Likewise, the secondary forest would increase from 57.7% (2017) to 73%. For this scenario, the changes in human settlement and water bodies would not increase or reduce their area. Conversely, the rate of change of primary forest and secondary forest were the greatest between 2017 and 2050. Regarding the optimistic scenario, it showed reductions in areas of primary forest; however, in lower magnitudes than for the stationary and pessimistic scenarios. For the pessimistic scenario, the Markov matrix was modified considering a greater pressure on the forest ecosystem. The area without vegetation showed a positive trend, with 4.8% in 2017 and an increase to almost 8% in 2050. The secondary forest would go from 57.7% to 85.6% in 2050. Finally, the primary forest would reduce its area to a 8% and isolated forest areas would appear. The rate of change for this scenario were the ones that showed the highest values. The LULCC dynamics projected for 2050 for the three scenarios (stationary, optimistic, pessimistic) is presented in Table 9.

## DISCUSSION

In this study, scenarios of LULCC for 2017 and 2050 were generated for a temperate forest region of Chihuahua Mexico. The scenarios were developed in Dinamica-EGO. Results were consistent with the results described by *Maeda et al. (2011)*. For the generation of transitions and simulation of scenarios, LULC of 1990, 2005 and 2017 were determined. In general, proximity to sources with anthropogenic activity as well as topography were important factors influencing the change in forest cover. The exchange between primary forest and secondary forest represented the main transition between 1990 and 2017. This transition produced the greatest impact, in agreement with the results reported by *Pérez-Vega, Rocha Álvarez & Regil García (2016)*. Such transition was influenced by the altitude, slope, and density of water streams, in agreement with the results of *Armenteras et al. (2006)* and *Chadid et al. (2015)*. The transition from primary to secondary forest could be attributed to the reduction in pine vegetation, where shrubs would become dominant. A consequence of the reduction of primary forest is the migration of fauna, which deals with the dispersal of the seeds of large-crowned trees (*Lehouck et al., 2009*). Other consequences include the change of lands to livestock production systems (*Maeda et al., 2010a*; *Maeda et al., 2010b*) and the presence of areas with high solar incidence and low coverage, which are prone to fires deRezendeetal2015. Another reason for the

reduction of primary forest is the proximity to urban rural localities and roads, which is in agreement with the results reported by *Aguiar, Câmara & Escada (2007)* and *Osorio et al. (2015)*. The proximity to urban and rural communities indicates the possible extraction of wood for export and also facilitates the expansion of the agricultural or grazing frontier (*Chadid et al., 2015*). This can be verified by the number of sawmills in the study area. The process of deforestation/degradation is strongly related to this cause. In the forested areas of Chihuahua, the rural localities are in a high degree of marginalization (*González, 2012*) where there exist agricultural incentives PEF 2025 (*CONAFOR, 2001*), causing the possible increase of the areas without vegetation. Another reason for the degradation may be the distance to the main roads and the topographic position.

The results obtained for the different scenarios showed differences among the surfaces of land use. The stationary scenario resulted in a considerable change in the primary forest, mainly. This scenario considers that the transition values among land use coverages will continue. The long-term impacts of the deforestation/degradation process include increased reservoir sedimentation and decreased flows in the dry season (*Gingrich, 1993*). Although the optimistic scenario showed increases in non-forested areas, this scenario was the one that showed the greatest resistance fo the transitions from primary forest to any other LULC. This scenario considers the strict application of the regulation of forest resources, in agreement with the general trend in the protection of forest ecosystems to degradation (*UN, 2015*) and the projections of the PEF 2025 (*CONAFOR, 2001*). The pessimistic scenario showed the greatest losses in the coverage of the primary forest. In addition, the increase in areas without vegetation, which is mainly associated to cropping and the proximity to water currents, is one of the main outputs of the pessimistic scenario, which agrees with the study by *Elz et al. (2015)*. The increase in agricultural areas resulting from this scenario may benefit the inhabitants economically; however, the expansion of this type of land use/land cover could lead to a greater demand of water for irrigation purposes, which could potentially impact water resources (*Maeda et al., 2010a*; *Maeda et al., 2010b*).

Population growth (*Barni, Fearnside & de Alencastro Graça, 2015*), the market demand and the lack of technification for wood processing cause the opening of land and the extraction of wood for self-consumption. Taking these aspects into account, the simulation of changes in forest cover indicates pressure on forest resources, which is consistent with that found by *Kamusoko et al. (2011)*. As a consequence, forest degradation could lead to soil loss (*Quan et al., 2011*), loss in biodiversity (*Falcucci, Maiorano & Boitani, 2007*) and landscape connectivity (*Tambosi et al., 2014*), habitat fragmentation (*Nagendra, Munroe & Southworth, 2004*), the presence of invasive species (*Mas, Pérez-Vega & Clarke, 2012*), among others.

The LULCC model of this study incorporated the Markov chains, Cellular Automata and WoE methods. Several transitions were simulated as in the studies by *Soares-Filho et al. (2010)*, *Ferreira et al. (2013)* and *Elz et al. (2015)*. The validation was carried out based on the FSI, as it was also performed in previous research (*Ximenes et al., 2011*). The result of this analysis, where the three aforementioned methods are combined, highlighted the variables driving the process of degradation/deforestation, as well as the manipulation based on the

knowledge of the transition probabilities, being more suitable for the simulation of LULCC (*Mas & Flamenco, 2011*). The transition probability matrices revealed that the primary forest has a negative trend in its occupied area, suggesting that degradation will continue over this land use, this area of primary forest changed to secondary forest. Although the other transitions did not produce important changes in the spatial configuration of the landscape, but their cumulative long-term effect could negatively impact the functioning of the ecosystems and their biodiversity (*Pompa, 2008*).

In this study, we focused on hypothetical scenarios where the pressure of forest resources was controlled by changing the transition probability. However, it is necessary to study scenarios where market demand (*Merry et al., 2009*) or illegal timber extraction (*Chadid et al., 2015*) is considered. The wood clandestinage corresponds to 30% in the some forest management units of Chihuahua (*Silva, 2009*).

The scenarios are not exact projections of the future state of the environment (*Feng & Liu, 2012*). However, it is an alternative means of supporting forest managers, which can serve as a valuable tool for studying political decisions (*Kolb & Galicia, 2018*). That would lead to a better knowledge of forest exploitation and protection. Managers can take into account the proposed scenarios and take decisions based on the one with the most promising results.

Due to the distribution of economic information (municipality based) and the lack of information from georeferenced illicit extractions, we believe an approach such as agent-based models would help to improve the study and address these issues. Finally, the model did not consider climatic variations such as precipitation and temperature, which can affect patterns and dynamics in recovery zones. That should be implemented in future studies.

## CONCLUSIONS

The use of scenarios as a methodology to study LULCC has been studied in depth at different scales and in different areas. However, several improvements can be implemented. This study presents an approach that integrates expert knowledge, and geospatial technologies such as geographic information systems and spatial simulation. The developed scenarios were based on the application of the forestry law (non-spatially) as well as the state of the landscape, and not only on the extrapolation of past trends. In addition, the scenarios are spatially explicit, which allow identifying the spatial pattern of change and the possible critical areas of change in forest cover. Finally, this study contributes to the understanding of the future fragmentation of the forest cover. Therefore, the current decisions in the field of forest management and land use/land cover influence the future of our forests and can probably be represented in one of the three proposed scenarios.

### Funding
Consejo Nacional de Ciencia y Tecnología, Mexico provided the scholarship given to Jesús A. Prieto-Amparán to pursue his graduate studies. The funders had no role in study design, data collection and analysis, decision to publish, or preparation of the manuscript.

### Grant Disclosures
The following grant information was disclosed by the authors:
Consejo Nacional de Ciencia y Tecnología, Mexico.

### Competing Interests
The authors declare there are no competing interests.

### Author Contributions
- Jesús A. Prieto-Amparán conceived and designed the experiments, analyzed the data, contributed reagents/materials/analysis tools, prepared figures and/or tables, approved the final draft.
- Federico Villarreal-Guerrero conceived and designed the experiments, analyzed the data, authored or reviewed drafts of the paper, approved the final draft.
- Martin Martínez-Salvador and Carlos Manjarrez-Domínguez performed the experiments.
- Griselda Vázquez-Quintero performed the experiments, prepared figures and/or tables.
- Alfredo Pinedo-Alvarez conceived and designed the experiments, analyzed the data, contributed reagents/materials/analysis tools, authored or reviewed drafts of the paper, approved the final draft.

### Data Availability
Prieto Amparan, Jesus Alejandro (2019): rawdata. figshare. Figure. https://doi.org/10.6084/m9.figshare.7845212.v1.

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
