# Peer review of "Spatial near future modeling of land use and land cover changes in the temperate forests of Mexico"

_PeerJ, doi:10.7717/peerj.6617_

## Round 0.1 · original submission · Major Revisions

· Academic Editor

Major Revisions

I have now received two independent reviews of your paper. Both reviewers suggested major revisions as they noted that the paper is worthy of publication but requires some changes - clarity in methods and rewrite conclusion. Please note all the suggested changes from the two reviewers and carefully address them in your submission should you chose to submit after major revision of your paper. I hope it will improve the paper.

Reviewer 1 ·

Basic reporting

The manuscript titled “Spatial near future modeling of land use and land cover changes in the temperate forests of Mexico” aims to present hypothetical scenarios based on spatial modeling. While this study does not present a new approach as claimed in the conclusion section (line 379), it can still provide some useful information that are particularly relevant for local/state-level policymakers. In my opinion this study can use a stronger study design, and a thorough editorial check for improving the language. The manuscript is full of typographical errors (see examples below) and lacks important details in methods. Please see my detailed comments below.

Abstract:
1. The authors might consider starting this section with the big question/study rationale.
2. There is no mention of the study area/country here which should be included in the revised version.
3. Last sentence: ‘on’ is missing after ‘Based’.

Introduction:
1. The flow between the paragraphs are not evident. The first paragraph talks about forest ecosystems and drivers of deforestation (while not including the wealth of literature on these topics – a serious weakness in the manuscript), and the second paragraph jumps right into Mexico and temperate forests. There should be a couple of transitional sentences that describe importance and current state of temperate forest worldwide, and then Mexico.
2. Subsequent paragraphs discuss other studies that have created similar scenarios in other regions (again, supporting my view that this is not a new approach as claimed by the authors). However, the authors should have described the rationale for this study more broadly and how they achieved the study objectives. Then they should have defended their choices of specific methods/models with appropriate citations. Section starting at line 79 does not make any sense. Overall, this section lacks the structure required to set the stage for this work.

Typographical errors/language:
1. Dinamica EGO is consistently misspelled – abstract, lines 168, 201.
2. Line 97: ‘Specifically’, not ‘Especifically’
3. Line 97: ‘pessimistic’, not ‘pesimistic’
4. Line 133: ‘composites’, not ‘compositions’
5. Line 138: ‘where’, not ‘Where’
6. Line 190: ‘optimistic’, not ‘optimist’
7. Line 290: ‘greatest’ not ‘greateest’
8. Capitalization not required for the land cover class names (lines 332, 335, 336: Primary forest, Areas without vegetation); or they should use those words within quotation marks.

Experimental design

Materials and methods:
1. Data source: lines 124-130 are not really required, as this is pretty standard for any satellite remote sensing study.
2. Land use classification (should really be ‘land cover and land use classification’): the authors mentioned that they used Support Vector Machine (SVM) classification while they said they used the R package for random forest, which is an entirely different classification scheme. Unless this is an honest mistake, this would make me very concerned about the technical background of the team. If they insist that they used random forest package for SVM, I would like to see the R codes that they used.
3. Selection of variables and transitions: then they directly jumped to this section without mentioning anything about the models. I’m assuming this section refers to the data required for the model, but then we need to first understand how the model works and what is required.
4. Exploratory analysis of data: did they use the threshold of 0.5 (lines 181-183)? Then clearly mention here. Line 166: how do they ‘previously know’?
5. Simulation of the land use changes: the authors classified 1990, 2005 and 2017 images, but it is not clear from this section which ones were used for simulation.
6. Validation: this section describes model validation, but does not mention anything about land cover classification validation. In the results section (line 223), accuracy numbers are provided, but they need to describe exactly how they performed the accuracy assessment. Also, for the model validation, I’m not sure if 2005 was used (seems like the case from the figure captions). This section, like many others, requires a complete rewrite.
7. I’m also not sure how/why they chose 2050. It would make more sense to me to use 1990-2005 transition probabilities to simulate 2017 with three scenarios, include a comparison of observed 2017 and simulated 2017, and then continue with the observed trend for another year, say 2050 (with proper justification). It’s not clear to me why they chose 2005 to compare the observed vs. simulated (Fig. 2). If they have strong reasons, they need to make it clear in the manuscript. But then they mention doing just that in the results section (line 264).

Validity of the findings

Discussion/Conclusion:
1. The last paragraph ends abruptly. I think they can move parts of the conclusion (lines 367-376) to discussion.
2. Line 379: this is first time I’m learning that the authors integrated interviews and GIS, no mention of interviews until now.

·

Basic reporting

Overall a good paper with regard to literature, background, professional article and figures, etc. It needs additional work on English grammar and language editing. Minor details below:

ABSTRACT
Abstract should report the classification validation results-- specifically give the Kappa coefficient.
“For the study, a steady, an optimistic and a pessimistic scenario was proposed” should read “were.”
Primary forest should not be capitalized in the middle of a sentence. Alternatively, all land use types should be capitalized as in the next sentence they authors say “Secondary Forest.”
Use consistant naming of scenarios instead of switching from “steady” to “static,” which are equivalent but could confuse readers.
“primary forest got reduced” should read “was reduced” and clarify what the % refers to—the abstract does not name the study area. All of Mexico?

INTRODUCTION
Line 47-51 check and revise grammar: “Temperate forests of Mexico occupy 17% of the national territory, represented by 32 millions hectares. In this region, the greatest association of pine and oak forests in the world occurs… However, 40 thousand hectares of forests get on average lost annually.”
Line 77: Suggest using a citation relevant to this study like Galford et al. 2015 that used Bayesian WoE for scenarios evaluating policy plans for agriculture and forest in DRC.
Line 80, 83: “celular" should read “cellular”
Line 83: Cite also Soares-Filho et al. 2013 J. Environ. Sftwr. DOI: 10.1016/j.envsoft.2013.01.010.
Line 97: “Especifically” is not a word in English
Line 100 “extemsion” is not a work in English

DATA SOURCE
Line 123 Table 1 seems incomplete
Study area is not well defined. Please provide a description of why this site was chosen.

MODEL VALIDATION
Line 265: Fuzzy similiarity index (FSI) was previously defined so it can just say “FSI” here
Line 266: “DSI” should read “FSI”?

DISCUSSION
Lines 302-305 reads like a summary of the work; rather it should discuss the findings.

Table 5. What does “exchange” rates mean? Rate of change? Including numbers (ha or km2) could be useful in addition to the percentages.
Table 6. Unclear from the table caption and the table itself—what are these numbers? Units?
Table 9 caption should read “projected” instead of “proyected” scenarios.

Experimental design

This primary research examines scenarios of land use change for a study site in Mexico through 2050. The land use change model is parameterized from historical land use change maps. The methods are rigorous and described in detail. Things like the Dinamica-EGO transition matrix are provided, increasing transparency and detail useful to the reader.

Validity of the findings

The Dinamica-EGO platform is utilized for spatial simulation models with Bayesian Weights of Evidence. It is a strong method that is appropriately applied by the authors. The authors use fuzzy similarity index to validate their model, a method that shows good agreement between their simulations and observed data with transparency to the readers.
The conclusions could be more strongly stated instead of reviewing the work that was done.

---

## Round 0.2 · Minor Revisions

· Academic Editor

Minor Revisions

Thank you for revising the manuscript in response to two reviewers comments. Please note there are some minor revisions suggested by one reviewer. Please address them as soon as you can and resubmit the manuscript for consideration.

·

Basic reporting

Be consistent with "Weights of Evidence" or "weights of evidence." Both notations are used but pick one and use throughout-- I believe "Weights of Evidence" is standard.

Similarly, "Bayesian" should have a capital B each time it is used.

For references like GloVis, citation should be "USGS 2018" and the reference should list the website address. Do no give website addresses in the body of the paper.

In the Fuzzy Similarity Index figure (4), please make the x-axis units of distance, not pixels. One by one radius of a 30 m pixel would be 15 meters; 3x3 pixels would be 60 meters.

Experimental design

Excellent.

Validity of the findings

Well done.

Additional comments

Nicely improved in this version.

---

## Round 0.3 · accepted · Accept

· Academic Editor

Accept

Thank you for your patience and revising the manuscript so diligently. Your paper is now accepted for publication. Congratulations once again and thank you for choosing PeerJ as an outlet of your research work.